# Maternal Docosahexaenoic Acid Exposure Needed to Achieve Maternal–Newborn EQ

**DOI:** 10.3390/nu14163300

**Published:** 2022-08-12

**Authors:** Danielle N. Christifano, Kathleen M. Gustafson, Susan E. Carlson, Nasrin Sultanna, Alexandra Brown, Scott A. Sands, John Colombo, Byron J. Gajewski

**Affiliations:** 1Department of Dietetics and Nutrition, University of Kansas Medical Center, Kansas City, KS 66160, USA; 2Hoglund Biomedical Imaging Center, University of Kansas Medical Center, Kansas City, KS 66160, USA; 3Department of Neurology, University of Kansas Medical Center, Kansas City, KS 66160, USA; 4Department of Biostatistics & Data Science, University of Kansas Medical Center, Kansas City, KS 66160, USA; 5Department of Psychology, Schiefelbusch Institute for Life Span Studies, University of Kansas, Lawrence, KS 66045, USA

**Keywords:** docosahexaenoic acid, pregnancy, maternal–newborn DHA EQ

## Abstract

Achieving maternal docosahexaenoic acid (DHA) status equal to or greater than the infant’s DHA status at delivery is known as maternal–newborn DHA equilibrium (EQ) and is thought to be important for optimizing newborn DHA status throughout infancy. The objective of this study was to determine the daily DHA intake during pregnancy most likely to result in EQ. The participants (*n* = 1145) were from two randomized control trials of DHA supplementation in pregnancy. DHA intake was estimated using an abbreviated food frequency questionnaire. Total DHA exposure during pregnancy was calculated as a weighted average of the estimated DHA intake throughout pregnancy and the randomized DHA dose (200, 800, 1000 mg). Red blood cell DHA was measured from maternal and cord blood plasma at delivery and EQ status was calculated. The DHA intake required to achieve EQ was estimated by regression. In terms of DHA exposure, the point estimate and 95% confidence interval to achieve EQ was 643 (583, 735) mg of DHA/day. The results of our trial suggest an intake of 650 mg of DHA/day is necessary to increase the potential for EQ at delivery. The clinical benefits of achieving EQ deserves continued study.

## 1. Introduction

Docosahexaenoic acid (DHA) is an omega 3 fatty acid that is increasingly recognized as being important for optimal maternal health and fetal/newborn development. While DHA can be synthesized from α-linolenic acid (ALA), the percentage of ALA converted to DHA is low and insufficient to meet the DHA (the richest source being fatty ocean fish) requirements in pregnancy necessary to achieve optimal levels for numerous physiological functions. As pregnant women living in the United States report low levels of seafood consumption, they must therefore rely on supplementation to attain optimal intake [1] The National Academy of Medicine (previously known as the Institute of Medicine) currently does not provide a recommended intake of DHA during pregnancy, despite mounting research supporting DHA supplementation as a strategy to improve birth outcomes, including the reduction of risk for preterm birth [2].

A concept put forth by Kuipers et al. [3] and Luxwolda et al. [4] suggests that achieving maternal DHA status equal to or greater than the infant’s DHA status at delivery (i.e., maternal–newborn DHA equilibrium) is important for optimizing newborn DHA status throughout infancy. DHA equilibrium (EQ) is specifically defined as cord blood DHA status being equal to or less than maternal DHA status at delivery. Similar to most nutrients, the transfer of DHA from mother to fetus is dependent on maternal DHA status during pregnancy [5]. If EQ is achieved, the mother is thought to have sufficiently transferred adequate DHA to the fetus during gestation, leaving her with enough reserves to provide DHA through lactation to her infant [3].

In our recent clinical trial [6], women randomized to consume 800 mg of DHA/day during pregnancy were more likely to achieve EQ at delivery when compared to those randomized to 200 mg DHA/day. EQ never occurred among women with low DHA status at delivery (defined as red blood cell phospholipid DHA, or RBC-PL-DHA, below 6.96% of total fatty acids) [6]. However, this trial did not identify the amount of DHA a pregnant woman should consume through diet and supplements to increase the potential for EQ.

To address this gap, we leveraged data from two large clinical trials of DHA supplementation in pregnant women (Prenatal Autonomic Neurodevelopmental Assessment (PANDA): NCT02709239; and Assessment of DHA on Reducing Early Preterm Birth (ADORE): NCT02626299) to predict the DHA exposure most likely to result in maternal–newborn EQ—an endeavor that can guide recommendations for maternal DHA intake during pregnancy.

## 2. Subjects and Methods

### 2.1. Subjects

The data for this analysis come from two recently completed randomized control trials: PANDA and ADORE. Both trials were registered in ClinicalTrials.gov and primary outcomes have been published [6,7]. Women who participated in the PANDA and ADORE trials were age 18 or older with singleton pregnancies and were English or Spanish speaking (ADORE only). At the baseline study visit (12–20 weeks gestation), women were randomized to receive a low dose of DHA (200 mg/day) or a high dose of DHA (800 mg/day: PANDA; 1000 mg/day: ADORE) for the duration of their pregnancy.

Participants were included in this analysis (*n* = 1145) if they provided dietary intake data at baseline and had maternal and cord blood RBC-PL-DHA levels at delivery. CONSORT diagrams and demographics for each parent trial are available in the primary publications [6,7].

### 2.2. DHA Exposure

DHA intake in early pregnancy was estimated using an abbreviated food frequency questionnaire (DHA-FFQ). The first 6 questions of the DHA-FFQ assess consumption of DHA-rich foods (3 questions ask about seafood in different categories of DHA content and the next 3 questions as about eggs, poultry, and liver). A 7th question asks about DHA supplement intake in the past two months [8]. Intake of DHA using the DHA-FFQ is a predictor of RBC-PL-DHA in a pregnant populations [9] and is feasible for use in a clinical setting [10]. Participants completed the DHA-FFQ questionnaire through an interview with trained research staff at the time of enrollment.

Total DHA exposure during pregnancy was calculated for each participant as a weighted average of the estimated DHA intake in early pregnancy and the estimated DHA intake after enrollment from the diet and the DHA dose the participant was randomized to during the clinical trial (200, 800, 1000 mg). DHA intake before enrollment (DHA_early_) was determined from the answers to all 7 questions of the DHA FFQ. DHA intake after enrollment (DHA_late_) included answers to the first six questions from the DHQ-FFQ to represent usual dietary DHA intake and the DHA dose assigned (Dose). The calculation for DHA_late_ does not include question 7 (DHA intake from supplements) since participants were asked to stop personal supplementation at enrollment and take the DHA supplements provided through the study. The weight (w) = (gestational age at enrollment)/37, where 37 represents the number of weeks of a gestation at term. Specifically, the formula for $DHA_{intake}=w*DHA_{early}+\left(1-w\right)*DHA_{late}$. Figure 1 is an example calculation for a participant randomized to 200 mg DHA/day.

### 2.3. DHA Status

RBC-PL-DHA was measured at baseline and at delivery. Blood samples were collected by venipuncture and placed on ice, centrifuged within 24 h to separate plasma, buffy coat, and anticoagulated RBCs, and stored at −80 °C until analysis. The lipids of the RBCs were extracted, phospholipids were separated by thin layer chromatography, fatty acids were transmethylated with boron trifluoride, and the fatty acid methyl esters were separated by gas chromatography according to methods that were previously published [7,11]. RBC-PL-DHA is reported as weight percentage of total phospholipid fatty acids.

### 2.4. Ethics

Written consent was obtained for all subjects and both trials were approved by the Institutional Review Board—Human Subjects Committee at the University of Kansas Medical Center (STUDY00003455 and STUDY00003792).

### 2.5. Statistics

The DHA intake required to achieve EQEQ was estimated by regressing EQEQ, calculated as $d=RBC.DHA_{M}-RBC.DHA_{CB}\;$on DHA_intake_; M refers to maternal RBC-PL-DHA at delivery, and CB refers to infant cord blood RBC-PL-DHA at delivery. This regression also adjusted for mean centered pre-pregnancy BMI, age at enrollment, and diagnosis of gestational diabetes (GDM) as covariates, because of their possible influence on the placental transfer of nutrients such as DHA. The regression equation is written as $d=\beta_{0}+\beta_{1}*DHA_{intake}+\overset{4}{\underset{j=2}{\sum }}(\beta_{j}*centered\;covariates_{j})+error;\;j=2,\;3,\;4$. The value of the DHA intake that results in EQEQ is—β_0_/β_1_. Using a bootstrap algorithm, we calculated a point estimate and 95% confidence interval of the value of this DHA intake that results in EQEQ.

## 3. Results

In the study sample (*n* = 1145), maternal–newborn DHA EQEQ was achieved by 21.9% of dyads in the 200 mg group and 52.8% of dyads in the 800 mg/1000 mg groups. According to questions 1–6 from the DHA-FFQ, the mean DHA intake from diet was 88 mg/day at the time of enrollment. Total DHA exposure was estimated to be 161 mg/day prior to enrollment (DHA_early_) when personal DHA supplementation prior to the trial was included. The average DHA intake after enrollment and randomization to supplementation (DHA_late_) was 682 mg/day.

The summary statistics for all variables in the regression are displayed in Table 1. GDM diagnosis was removed from the regression because a one-way ANOVA indicated insignificant results (F = 1.013, *p* = 0.314). Final regression results are shown in Table 2. In terms of DHA exposure, the point estimate and 95% confidence intervals to achieve EQEQ were 643 (583, 735) mg of DHA/day. The RBC-PL-DHA level at which EQEQ occurred was 10% (Figure 2).

## 4. Discussion

The primary goal of this study was to determine the amount of DHA intake required to achieve maternal–newborn DHA EQ using information from a large sample of pregnant women. Using a combination of dietary DHA intake and supplement intake during pregnancy, we estimate the optimal DHA intake for most women to achieve EQ is close to 650 mg/day. Prior to this, we knew only that the women randomized to receive 800 mg of DHA/day were more likely to achieve EQ when compared to women who received 200 mg of DHA/day [6].

As noted previously, maternal–newborn DHA EQ is the theoretical point at which a mother’s RBC-DHA level at delivery is sufficient to prevent a decline in maternal DHA status during lactation [3]. Without supplementation, maternal DHA status decreases from delivery to 3 months postpartum, with greater declines evident in women with a lower intake of seafood [3,12,13]. The DHA content of breastmilk is related to maternal dietary intake of DHA [14]. Women who achieve EQ have higher amounts of DHA in their breastmilk and are less likely to have a decrease in maternal DHA status during lactation [3].

Through this analysis, we also determined the RBC-PL-DHA at which EQ occurs. In a Tanzanian cohort with varying levels of seafood intake, maternal–infant EQ occurred when maternal RBC-DHA reached 6.1 g% [4]. Although we report a level of 10% RBC-PL-DHA here, it is important to remember that methods for measuring DHA status vary among studies. At the same time all methods for measuring DHA status from whole blood and various blood compartments are highly correlated [15]. Because of this, we place the most emphasis on the DHA intake required to reach EQ as it has the potential to easily translate to clinical recommendations and is universal to all DHA research.

Current DHA intake recommendations in pregnancy vary among recommending bodies in the United States. The American College of Gynecology (ACOG) recommends at least two servings (one serving = 8–12 oz) of fish or shellfish per week before pregnancy, during pregnancy, and while breastfeeding. The Dietary Guidelines for Americans 2020–2025 recommend 8–12 oz of seafood per week during pregnancy [16]. The US Food and Drug Administration and Environmental Protection Agency also recommends two to three servings of seafood rich in omega 3 fatty acids per week for the general population. Expert committees and agencies including the March of Dimes [17], World Association of Perinatal Medicine [18], and others [19] recommend 200–300 mg of DHA per day during pregnancy as a dose that could possibly result in benefit and carries very low risk of harm.

The results of our trial suggest an intake of 650 mg DHA/day—a dose more than twice what is currently recommended—is necessary to increase the potential for EQ EQat delivery. According to the USDA database, FoodData Central [20], a 3 oz portion of salmon contains approximately 1200 mg of DHA, while other fatty fish such as herring, sardines, mackerel, and trout contain 900, 700, 600 and 400 mg of DHA per 3 oz serving, respectively. Shellfish such as oysters, shrimp, and scallops contain much lower quantities of DHA per serving (100–200 mg per 3 oz serving). To consume the equivalent of 650 mg of DHA/day (4550 mg/week), a pregnant woman would have to consume nearly four 3-oz servings of salmon, eleven 3-oz servings of trout, or forty-five 3-oz servings of shellfish each week. While seafood recommendations are valuable for some, most pregnant women do not consume seafood regularly as indicated by NHANES data from 2001–2014, where the mean intake of seafood of women of childbearing age was less than 0.5 oz per day [1]. When taking into account the current dietary trends of low seafood consumption, supplementation of DHA is necessary in meeting intake goals of 650 mg of DHA/day.

DHA is preferentially transferred across the placenta [5] and transfer appears to be at the expense of the mother’s DHA status until EQ is reached [6]. As evidenced in Figure 2, maternal and fetal RBC-DHA levels increase throughout gestation. However, the slope of the rise in fetal RBC-DHA throughout gestation is greater than the slope of the rise in maternal RBC-DHA because of the preferential transfer of DHA to the fetus to support multiple aspects of fetal growth and child development. For example, we observed that young children of DHA supplemented women did not experience the same increase in diastolic and systolic blood pressure with overweightness as their peers whose mothers were not supplemented [21]. The DHA status of the offspring is further enhanced by receiving milk from a mother who achieved EQ during her pregnancy as shown by Kuipers et al. [3]. Additionally, postnatal supplementation with DHA has been shown to improve cortical visual acuity and cognitive function [22,23]. While EQ is an indication of the successful placental transfer of DHA, EQ does not fully capture the complexities involved in fetal uptake of this important nutrient. The mechanisms underlying the transfer and metabolism of fetal DHA are not fully understood; however, elucidating the lipid forms utilized by the fetus (i.e., DHA-phosphatidyl-ethanolamine vs. DHA-phosphatidylcholine) [24], the role of specific phospholipases, and the expression of genes (e.g., MFSD2a [25]) are necessary to fully appreciate the value of EQ.

Whether or not the improvement in maternal DHA status that accompanies EQEQ produces advantages for the mother herself is an area that deserves study. Certainly, there is evidence that maternal DHA supplementation has advantages in terms of reduced rates of pre-eclampsia during pregnancy [26], and supplementation increases the chances of achieving EQEQ, however, it is not known if EQ is necessary to achieve these benefits. Whether or not EQEQ is correlated with any clinical outcomes is unknown and future work should aim to elucidate this relationship.

## Figures and Tables

**Figure 1 nutrients-14-03300-f001:**
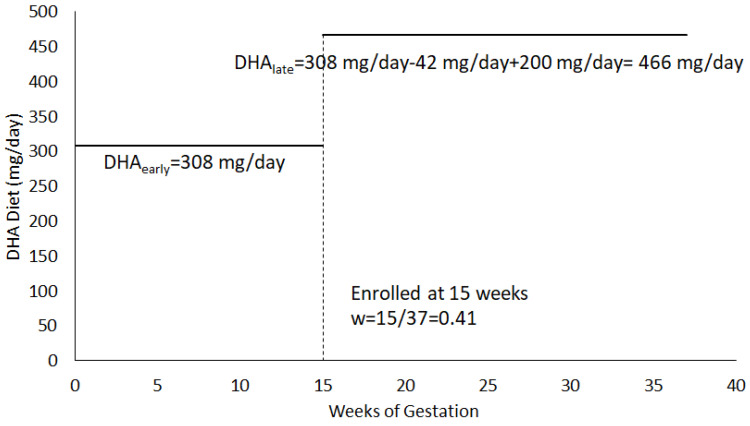
Example DHA Exposure Calculation for a participant randomized to 200 mg/day. For this example, $DHA_{intake}=0.41*308+\left(1-0.41\right)*466=401.22\;\mathrm{mg}/\mathrm{day}$.

**Figure 2 nutrients-14-03300-f002:**
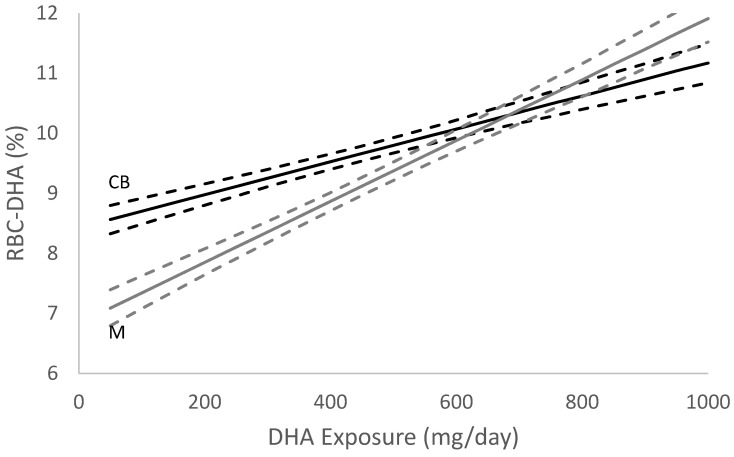
DHA Equilibrium (EQ) Estimate (*n* = 1145). Graphical representation of the EQEQ location with point estimates and 95% confidence intervals for DHA exposure. RBC-DHA is the red blood cell DHA %, dark line is the regression estimate of cord blood RBC-DHA (CB) and grey line is the regression estimate of maternal RBC-DHA (M) as a function of DHA exposure. The dashed lines are regression 95% intervals.

**Table 1 nutrients-14-03300-t001:** Summary Statistics (mean, median, and standard deviation for all variables).

	Mean	Median	Standard Deviation
DHA intake from diet (DHA_intake_l) (mg/day)	88.3	70.0	79.5
DHA exposure in early pregnancy including diet and supplements (DHA_early_) (mg/day)	160.8	122.0	135.0
DHA exposure in late pregnancy (DHA_late_) (mg/day)	682.3	842.0	387.3
Total DHA exposure during pregnancy (mg/day)	445.5	487.9	228.5
Pre-pregnancy BMI (kg/m^2^)	27.9	26.5	7.0
Age at enrollment (years)	30.2	30.2	5.5
GDM (n (%) diagnosed)	11.3%		

**Table 2 nutrients-14-03300-t002:** Regression of $d=RBCDHA_{M}-RBCDHA_{CB}$ vs. DHA exposure (mg/d) with centered BMI and centered age as covariates, where RBCDHA_M_ = maternal DHA at delivery and RBCDHA_CB_ = cord blood DHA at delivery.

	Estimate	Standard Error	t Value	*p*-Value
Intercept	−1.6544	0.1373	−12.051	<0.001
Total DHA Exposure	0.0026	0.003	9.346	<0.001
Centered Pre-Pregnancy BMI	−0.0442	0.089	−4.983	<0.001
Centered Age at Enrollment	0.0827	0.0116	7.121	<0.001

## Data Availability

We are willing to share deidentified data from the study with a signed data access agreement that includes the study principal investigators, contingent on approval of the planned use of the data. As the data are entered into an electronic system, a specific request to DNC (dchristifano@kumc.edu) would be needed to generate a data output for other investigators.

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
