# Peer review of "Maternal Docosahexaenoic Acid Exposure Needed to Achieve Maternal–Newborn EQ"

_nutrients, 2022, doi:10.3390/nu14163300_

Round 1
Reviewer 1 Report
The is a well written manuscript using existing data from two cohorts with varying levels of docosahexaenoic acid (DHA) supplementation during pregnancy to address the theory of a maternal-fetal DHA equilibrium. This is thought to be the level required to ensure the mother retains adequate DHA for lactation. The study presents data on DHA as a percent of total fatty acids in RBC phospholipids ( ie incorporated into tissue) obtained from mothers and cord blood at delivery. The data are adjusted for maternal DHA exposure from dietary and supplemental sources (200 to 1000 mg per day according to the prior studies protocol). Regression of the continuous variable of RBC DHA in maternal and cord blood with DHA exposure found an intersection at approximately 650 mg/day and the authors suggest this is the point at which maternal and fetal DHA status are equilibrated. What is missing from this analysis is the fact that the total lipids in the fetal and maternal circulation are not equal. The maternal levels of DHA are higher than in the fetus in all lipid classes, non esterified “free” fatty acids, triglycerides, phospholipids and cholesterol esters in terms of absolute concentrations. The data in this analysis demonstrates that incorporation of DHA into RBC phospholipids has reached a similar % of total, which likely suggests adequate DHA has reached the fetal compartment. But this data does not indicate that levels are equal on both sides of the placenta, higher maternal levels are maintained because transfer from mother to fetus will continue down the concentration gradient established in large part due to maternal hyperlipidemia of pregnancy.
While this data is useful and suggests that high maternal intake is needed for fetal RBC levels to reach a similar percentage as in the mother, the data is somewhat complicated by the fact that as maternal levels rise so do fetal levels. The intersection in Figure 2 is due to the fact that the two slopes are not the same with maternal levels rising more than the fetal levels for the same dietary intake. Fetal DHA RBC content (as a % of total lipids) is higher than maternal due to the preferential transfer of this critical fatty acid so the two lines intersect at a relatively high total maternal intake. It is not completely clear what biological impact this intersection has for fetal brain or adipose DHA levels, but one can assume they are likewise higher. Achieving a high percentage of DHA in RBC phospholipids in the maternal circulation will increase availability to the placenta for transport but the current understanding is that DHA is transported as a non-esterified fatty acid or perhaps a LysoPhosphatidyl Choline form. The role of phospholipases to release DHA from the phospholipid pool for transfer to the fetus remains unclear. The rate limiting step for fetal transfer and incorporation may be these aspects since placental MFSD2a expression has been shown to correlate well with fetal DHA levels.
I think the data are of value and quite interesting but exactly what they mean biologically for the developing fetus and for lactation should be discussed in a bit more detail. More DHA is better for sure but what exactly is happening at this equilibrium point is less clear.
Author Response
Dear Reviewer 1,
Thank you for your thoughtful review of our manuscript, “Maternal docosahexaenoic acid exposure needed to achieve maternal-newborn equilibrium”. You made several excellent points which we have addressed in our discussion. Specifically, we have included additional text highlighting the complexities of placental transfer and metabolism of DHA that are not captured in a crude measure of equilibrium. We agree with the reviewer that these complexities must be considered when determining the relative value of achieving equilibrium. The specific edits to the discussion are below in italicized text.
DHA is preferentially transferred across the placenta5 and transfer appears to be at the expense of the mother’s DHA status until equilibrium is reached.6 As evidenced in Figure 2, maternal and fetal RBC-DHA levels increase throughout gestation. However, the slope of the rise in fetal RBC-DHA throughout gestation is greater than the slope of the rise in maternal RBC-DHA because of the preferential transfer of DHA to the fetus to support multiple aspects of fetal growth and child development. For example, we observed that young children of DHA supplemented women did not experience the same increase in diastolic and systolic blood pressure with overweight as their peers whose mothers were not supplemented.21 DHA status of the offspring is further enhanced by receiving milk from a mother who achieved equilibrium during her pregnancy as shown by Kuipers et al.3 Additionally, postnatal supplementation with DHA has been shown to improve cortical visual acuity and cognitive function.22,23
While EQ is an indication of successful placental transfer of DHA, EQ does not fully capture the complexities involved in fetal uptake of this important nutrient. The mechanisms underlying the transfer and metabolism of fetal DHA are not fully understood; however, elucidating the lipid forms utilized by the fetus (i.e. DHA-phosphatidyl-ethanolamine vs. DHA-phosphatidylcholine)24, the role of specific phospholipases, and the expression of genes (e.g. MFSD2a25) are necessary to fully appreciate the value of EQ.
Thank you!
Reviewer 2 Report
Manuscript entitled „Maternal docosahexaenoic acid exposure needed to achieve maternal-newborn equilibrium” By Christifano et al. aims to identify the amount of DHA intake by pregnant women required to achieve maternal-newborn DHA equilibrium. Maternal–infant equilibrium occurs when cord blood DHA level is less than or equal to maternal DHA at delivery. Studies were performed on 1,145 women and their infants. DHA intake was estimated using a questionnaire what is some limitation, however justified in the case of such large human studies. Outcomes presented in proposed manuscript bring valuable knowledge in the field.
Minor changes:
- Abbreviation EQ for term “maternal-newborn DHA equilibrium” is used in the first sentence of the abstract. Later in the abstract word equilibrium is used again instead of its abbreviation
- Line 31, here it would be good to add information about the percentage of ALA that is converted to DHA, that will highlight the fact that organism is not able to synthesise DHA that would meet its demands
- Line 83, abbreviation (w) is used for the “weighted average”, while in line 91 it is used again to name weight related to time in which certain amounts of DHA were intaken. Please, be specific.
- Figure 1 if you decided to enclose figure explaining the example calculation of daily DHA intake it would be good to present in it also whole formula
- In discussion you said about two goals. In my opinion we can speak about one goal: estimation of EQ, as for its assignation essential is RBC-DHA (proposed second goal)
Author Response
Dear Reviewer 2,
Thank you for your review of our manuscript, “Maternal docosahexaenoic acid exposure needed to achieve maternal-newborn equilibrium”. You made several excellent points which we have addressed in italicized text below.
Minor changes:
- Abbreviation EQ for term “maternal-newborn DHA equilibrium” is used in the first sentence of the abstract. Later in the abstract word equilibrium is used again instead of its abbreviation
We now only use the EQ abbreviation in the abstract after the first mention.
- Line 31, here it would be good to add information about the percentage of ALA that is converted to DHA, that will highlight the fact that organism is not able to synthesise DHA that would meet its demands
We added a line to highlight the low rate of conversion of ALA to DHA.
- Line 83, abbreviation (w) is used for the “weighted average”, while in line 91 it is used again to name weight related to time in which certain amounts of DHA were intaken. Please, be specific.
We deleted the first reference to (w) to avoid confusion.
- Figure 1 if you decided to enclose figure explaining the example calculation of daily DHA intake it would be good to present in it also whole formula
The whole formula is presented in the text and the formula with a sample calculation has also been added to figure 1.
- In discussion you said about two goals. In my opinion we can speak about one goal: estimation of EQ, as for its assignation essential is RBC-DHA (proposed second goal)
Thank you for this point. We changed the language and deleted the reference to a “secondary goal”.
Thank you!